# Multi-Faceted Bioactivity Assessment of an Exopolysaccharide from *Limosilactobacillus fermentum* NCDC400: Antioxidant, Antibacterial, and Immunomodulatory Proficiencies

**DOI:** 10.3390/foods12193595

**Published:** 2023-09-27

**Authors:** Manorama Kumari, Basavaprabhu Haranahalli Nataraj, Writdhama G. Prasad, Syed Azmal Ali, Pradip V. Behare

**Affiliations:** 1Technofunctional Starters Lab, National Collection of Dairy Cultures (NCDC), Dairy Microbiology Division, ICAR-National Dairy Research Institute, Karnal 132001, India; 2Dairy Chemistry and Bacteriology Section, Southern Regional Station, ICAR-National Dairy Research Institute, Bengaluru 560030, India; 3Dairy Technology Division, ICAR-National Dairy Research Institute, Karnal 132001, India; writdhama_3993@rediffmail.com; 4Cell Biology and Proteomics Lab, Animal Biotechnology Center, ICAR-National Dairy Research Institute, Karnal 132001, India; 5Proteomics of Stem Cells and Cancer, German Cancer Research Center (DKFZ), 69121 Heidelberg, Germany

**Keywords:** *Limosilactobacillus fermentum*, exopolysaccharide, antioxidant, oxidative stress, antibacterial, immunomodulation

## Abstract

Exopolysaccharides (EPS) are acknowledged for their diverse functional and technological properties. This study presents the characterization of EPS400, an acidic exopolysaccharide sourced from the native probiotic *Limosilactobacillus fermentum* NCDC400. Notably, this strain has demonstrated previous capabilities in enhancing dairy food texture and displaying in vivo hypocholesterolemic activity. Our investigation aimed to unveil EPS400′s potential biological roles, encompassing antioxidant, antibacterial, and immunomodulatory activities. The results underscore EPS400′s prowess in scavenging radicals, including the 2,2-diphenyl-1-picrylhydrazyl radical, 2,2′-azino-di-(3-ethylbenzthiazoline sulfonic acid) radical, superoxide radical, hydroxyl radical, and chelating activity targeting the ferrous ion. Furthermore, EPS400 displayed substantial antibacterial effectiveness against prevalent food spoilage bacteria such as *Pseudomonas aeruginosa* NCDC105 and *Micrococcus luteus*. Remarkably, EPS400 exhibited the ability to modulate cytokine production, downregulating pro-inflammatory cytokines TNF-α, IL-1β, IL-6, and nitric oxide, while concurrently promoting the release of anti-inflammatory cytokine IL-10 within lipopolysaccharide-activated murine primary macrophages. Additionally, EPS400 significantly (*p* ≤ 0.05) enhanced the phagocytic potential of macrophages. Collectively, our findings spotlight EPS400 as a promising contender endowed with significant antioxidant, antibacterial, and immunomodulatory attributes. These characteristics propose EPS400 as a potential pharmaceutical or bioactive component, with potential applications in the realm of functional food development.

## 1. Introduction

Oxidative stress and inflammation play a pivotal role in the progression of a wide spectrum of metabolic disorders and chronic ailments in human beings. These conditions encompass diabetes mellitus, cancers, hypertension, coronary heart disease, and various other pathologies that also exacerbate unfavorable outcomes in cases of Coronavirus disease (COVID-19) [1,2]. The profound shift in dietary patterns towards meals with a high fat, high sugar, and high salt (HFSS) content, often at the expense of essential micronutrients and dietary fiber, is a prominent catalyst for oxidative stress and inflammation [3]. Efforts to mitigate oxidative stress and inflammation have involved the application of synthetic additives; however, this approach has led to a range of organ-related dysfunctions. As a result, the quest for natural therapeutic reservoirs has become a focal point for researchers in the food and pharmaceutical domains, with the intention of circumventing these adverse effects.

Natural bioactive compounds, exemplified by exopolysaccharides (EPSs), have emerged as a potential substitute for synthetic additives within the realms of both food and pharmaceuticals [4]. These substances are favored due to their inherent natural origins, safety profile, sustainable nature, economic relevance, and health-enhancing characteristics [5,6]. Noteworthy in this context is the EPS synthesized by lactic acid bacteria (LAB), which holds promise as a novel antibacterial, antioxidant, and immunopotentiating agent in the food and pharmaceutical sectors. Its utility extends to improving the technological attributes of food products, curbing the growth of foodborne pathogens, and augmenting health-related advantages [5,6,7]. EPS, derived from LAB, can be used as food additive or functional food ingredient in the production of a variety of foods, including fermented dairy foods, fermented meat products, plant-based yogurt-like beverages, and bakery products [8]. EPS can also be used as a prebiotic food supplement because it is selectively fermented by the intestinal microbiota when it reaches the colon undigested, conferring health benefits through the formation of short-chain fatty acids (SCFAs) [9].

However, it is important to note that a substantial portion of commercially available EPSs intended for food and pharmaceutical applications are derived from non-lactic-acid bacteria, exemplified by substances like xanthan gum, cellulose, and beta-glucan. This may be due to the lower yield of EPS derived from LAB [10]. Although the yield of EPS produced by LAB is limited, the exploration of novel EPS based on their unique qualities and potential biological activity would allow for their application in the food industry. Furthermore, the specific impact of antioxidants, coupled with the resulting immune-modulatory effects stemming from EPS sourced from LAB, remains an area with comparatively limited exploration.

Hence, the exploration of novel exopolysaccharides (EPS) originating from probiotic strains of *Limosilactobacillus fermentum* holds an intriguing allure. Specifically, *L. fermentum* NCDC400, isolated from the traditional Indian dairy product known as “dahi,” has emerged as a lactic strain of notable technological significance. This strain not only produces EPS, but also enhances the rheological and sensory attributes of low-fat fermented dairy products [7]. In addition to its technological prowess, this strain exhibits distinctive safe probiotic functionalities [11,12]. It has been observed to contribute to the maintenance of testosterone levels and the enhancement of sperm motility within a diet-induced obese animal model [13]. Moreover, *L. fermentum* NCDC400 showcases a preventive influence on obesity-related hepatic steatosis, coupled with improvements in antioxidant and anti-inflammatory traits in mice subjected to a high-fat diet [14]. Furthermore, different strains of *L. fermentum* have been reported to show anti-oxidant properties and other health-promoting properties that help control a variety of host diseases [15]. Freire et al. (2021) found that administering the *L. fermentum* formulation reduced low-grade inflammation and oxidative stress biomarkers in colon and heart tissues [16].

A prior investigation has encompassed the characterization of EPS400 derived from *L. fermentum* NCDC400. This particular EPS variant, labeled as EPS400, possesses a molecular weight of 5.7 × 10^5^ Da and assumes the form of a heteropolysaccharide. Its composition is defined by a ratio of glucose to rhamnose to galactose in the proportions of 1:13:1.5. The structural makeup incorporates a diverse array of functional groups, including hydroxyl (O-H) and carboxyl (C=O) moieties. Additionally, it manifests (1 → 4)-α-D-glucopyranose (Glcp) and (1 → 6)-α-D-glucopyranose (Glcp) linkages [5,7]. While EPS400 has demonstrated an array of health advantages, its potential antibacterial, antioxidant, and immunomodulatory capabilities warrant further exploration. Therefore, the principal objective of this study is to delve into the in vitro antioxidative and antibacterial activities exhibited by EPS400. Additionally, the study aims to assess the immunomodulatory potential of EPS400 in murine peritoneal macrophages activated by lipopolysaccharide (LPS). The outcomes of this investigation hold the potential to unveil a novel EPS variant endowed with antibacterial, antioxidant, and immunomodulatory attributes. Such a bioactive compound could find application as a key component in pharmaceuticals or functional foods, thus contributing to diverse health-promoting endeavors.

## 2. Material and Methods

### 2.1. Production and Extraction of Exopolysaccharide

EPS-producing *Limosilactobacillus fermentum* NCDC 400 strain (hereinafter, *L. fermentum* NCDC400), was obtained from the National Collection of Dairy Cultures (NCDC), ICAR-National Dairy Research Institute, Karnal, Haryana, India. The strain was cultured and propagated in sterile De Man, Rogosa, and Sharpe (MRS) broth (Himedia, Mumbai, India) at 42 °C for 18 h. The EPS was produced in a previously optimized deproteinized whey medium and extracted via the repetitive ethanol precipitation method [7]. The freeze-dried crude EPS was purified using anion exchange chromatography and lyophilized according to the method described by [5]. The lyophilized purified EPS was designated “EPS400”. The yield of EPS400 on a wet basis (g/L), on a dry basis (mg/L), and on a dry basis (mg/mg of the bacterial cell) was determined.

### 2.2. Determination of the Antioxidative Activity of EPS400

#### 2.2.1. Preparation of EPS400 Solution

The EPS solution was prepared by dissolving lyophilized EPS400 powder in Phosphate-Buffered Saline at concentrations of 0.125, 0.25, 0.5, 1.0, 2.0, 3.0, 4.0, and 5.0 mg/mL. (PBS, pH 7.2). The scavenging activity of EPS400 on DPPH, ABTS, hydroxyl, and superoxide radicals was evaluated using an ascorbic acid (VC) solution (Himedia, Mumbai, India) as a positive control. The chelating activity of EPS400 on ferrous ions was determined by comparing it with Ethylene diamine tetra acetic acid (EDTA) (Himedia, Mumbai, India).

#### 2.2.2. Determination of DPPH Scavenging Activity

The scavenging activity of EPS400 on 2,2-diphenyl-1-picrylhydrazyl (DPPH) free radical was measured according to the method of [17]. Briefly, 0.2 mM solution of DPPH (Sigma-Aldrich, Rockville, MD, USA) was prepared in 95% (*w*/*v*) ethanol and 2 mL of this solution was added to 1.0 mL of EPS400 (0.125 to 5.0 mg/mL). The mixture was shaken vigorously and kept for 30 min in the dark at room temperature. The absorbance was determined at 517 nm using a microplate reader (Agilent BioTeK, Santa Clara, CA, USA). The average percent of scavenging capacity was taken from three replicates. The scavenging activity of DPPH radical was calculated as follows:$$Scavenging\;activity\;\left(\%\right)=(1-\frac{Asample-Ablank}{Acontrol})\times 100$$
where *Asample* is the absorbance of the EPS400 mixed with DPPH-alcohol solution, *Acontrol* is the absorbance of the mixture containing only deionized water and DPPH solution, and *Ablank* is the absorbance of mixture containing only ethanol and EPS400.

#### 2.2.3. Determination of ABTS Scavenging Activity

The 2,2′-azino-di-(3-ethylbenzthiazoline sulfonic acid) (ABTS) radical-scavenging activity of EPS400 was determined according to the method of [18]. A stock solution was prepared by dissolving 7.4 mM ABTS (Sigma-Aldrich, USA) and 2.6 mM potassium persulfate (Sigma-Aldrich, USA) in deionized water. The concentrated ABTS stock solution was diluted with PBS and absorbance was recorded at 734 nm using a microplate reader (BioTeK, USA) after 16 h. Subsequently, 20 μL of the EPS400 (0.125–5.0 mg/mL) was mixed with 180 μL of the ABTS radical solution and the absorbance was measured. The percentage of scavenging activity of ABTS oxidation was calculated using the formula as follows:$$Scavenging\;activity\;\left(\%\right)=(1-\frac{Asample-Ablank}{Acontrol})\times 100$$
where *Asample* is the absorbance of the EPS400 mixed with ABTS solution, *Acontrol* is the absorbance of the mixture containing only deionized water and ABTS solution, *Ablank* is the absorbance of mixture containing only deionized water and EPS400.

#### 2.2.4. Determination of Hydroxyl Radical Scavenging Activity

The hydroxyl radical scavenging activity was analyzed according to the method of [17]. The reaction mixture contained 1 mL of 0.15 M sodium phosphate buffer (pH 7.4), 1 mL of 2.5 mM 1,10-phenanthroline (Himedia, Mumbai, India), 1 mL of 2.5 mM FeSO_4_ (Himedia, Mumbai, India), and 1.0 mL of EPS400 (0.125–5.0 mg/mL). The reaction started after adding 1 mL of 20 mM H_2_O_2_ (Sigma-Aldrich, USA), and the mixture was kept at 37 °C for 90 min. The absorbance was determined at 536 nm using a microplate reader (BioTeK, USA). The scavenging activity of hydroxyl radical was calculated as follows:$$Scavenging\;activity\;\left(\%\right)=(\frac{Asample-Ablank}{Acontrol-Ablank})\times 100$$
where *Asample* is the absorbance of the reagent mixture with the sample, *Ablank* is the absorbance of mixture without EPS400, and *Acontrol* is the absorbance of mixture without EPS400 and H_2_O_2_.

#### 2.2.5. Determination of Superoxide Anion Radical Scavenging Activity

The superoxide radical scavenging ability of EPS400 was assessed by the method reported by [17]. A total of 3.0 mL of 0.05 M Tris-HCl buffer (pH 8.2) (Sigma-Aldrich, USA) was mixed with 1.0 mL of EPS400 (0.125–5.0 mg/mL) and incubated at 37 °C for 10 min. After 10 min incubation, 40 μL of 45 mM pyrogallol (Himedia, India) was added to the mixture and kept for 5 min. The absorbance was measured at 320 nm using a microplate reader (BioTeK, USA).
$$Scavenging\;activity\;\left(\%\right)=(1-\frac{Asample}{Acontrol})\times 100$$
where *Asample* is the absorbance of the reagent mixture with EPS400, and *Acontrol* is the absorbance of the reagent mixture without the EPS400.

#### 2.2.6. Determination of Ferrous Ion Chelating Ability

Ferrous ion (Fe^2+^) chelating ability was investigated as previously described by [17]. The reaction mixture contained 1.0 mL of EPS400 (0.125–5.0 mg/mL), 0.05 mL of 2 mM ferrous chloride (FeCl_2_) solution (Himedia, India), 2 mL of 5 mM ferrozine solution (Himedia, India) and 2.75 mL deionized water, and was mixed well and incubated in a water bath at room temperature for 10 min. The absorbance of the mixture was measured at 562 nm against a blank (the same volume of PBS). The ferrous ion chelating ability was calculated as follows:$$Chelating\;ability\;\left(\%\right)=(Ablank-\frac{Asample-Ao}{Ablank})\times 100$$
where *Asample* is the absorbance of the reagent mixture with EPS400, *Ao* is the absorbance of the sample under identical conditions as a sample with deionized water instead of FeCl_2_ solution, and *Ablank* is the absorbance of the sample without EPS400.

### 2.3. Determination of Antimicrobial Activities of EPS400

Antimicrobial activity of EPS400 against bacterial (*Bacillus substilis* NCDC70, Micrococcus luteus NCDC174, *Pseudomonas aeruginosa* NCDC105, *Escherichia coli* NCDC135, *Staphylococcus aureus* NCDC100), and fungal species (*Rhodotorula glutinis* NCDC51, *Aspergillus niger* NCDC55, *Candida butyri* NCDC280 and *Penicillum camemberti* NCDC56) was determined using agar well diffusion assay [19] with some modification. Briefly, the overnight incubated indicator microorganisms were diluted to 10^6^ to 10^7^ cfu/mL and spread on a petri dish containing Luria–Bertani (Himedia, India) or potato dextrose agar (Himedia, India). Afterwards, 200 μL of EPS400 solution was poured into the wells (7 mm in diameter) created on the solid agar. The plates were incubated for 24 h and the inhibition zones were measured. Ampicillin (100 g/mL) (Himedia, India) served as a positive control.

### 2.4. Determination of Anti-Inflammatory Effects of EPS400 in Murine Peritoneal Macrophages

#### 2.4.1. Isolation and Preparation of Murine Peritoneal Macrophage

Eight male Swiss albino mice of 25–30 g were procured from the Small Animal House of National Dairy Research Institute (NDRI), Karnal, Haryana, India, approved by the Institutional Animal Ethics Committee (IAEC), National Dairy Research Institute, India (NDRI-IAEC Approval No. 46-IAEC-20-37 on dated 22 July 2020). The animals were sacrificed as humanely as possible using ether overdosing, and the peritoneal cavity macrophages were collected in RPMI-1640 (Himedia, India) supplemented with L-glutamine, 5000 U/mL penicillin, 5 mg/mL streptomycin (Himedia, India) following the method of [20]. The cell suspension was re-suspended in RPMI-1640 containing 10% fetal bovine serum (FBS) after being centrifuged at 350 g for 5 min at 4 °C. Using the trypan blue exclusion assay [21], the cell suspension was examined for the viability of macrophages using the following formula:$$Cell\;viability\left(\%\right)=\frac{Total\;viable\;cells\;(appearing\;colorless)}{Total\;counted\;cells}\times 100$$

#### 2.4.2. Measurement of Cell Viability by MTT Assay

The cytotoxic effect of lipopolysaccharide (LPS), as well as EPS400, on macrophage cells was tested using 3-(4,5-dimethylthiazol-2-yl)-2,5-diphenyltetrazolium bromide (MTT) assay [22]. Briefly, macrophages were seeded at a concentration of 2 × 10^5^ cells/well in 96-well plates (Thermo Scientific, Santa Clara, USA). After 2 h of incubation, varying concentrations of LPS (0.001, 0.01, 0.1, 1, and 10 µg/mL) (Sigma-Aldrich, USA) were added and incubated for 24 and 48 h to optimize the concentration of LPS, whereas varying concentrations of EPS400 (12.5, 25, 50,100, 200, 400, and 800 μg/mL) were added and incubated for 24 h to evaluate the cytotoxicity of the EPS400. Then, 10 μL of MTT dye (5 mg/mL concentration) (Himedia, India) and 90 μL RPMI medium were added to each well and incubated at 37 °C for an additional 4 h. The medium was removed using centrifugation at 3000× *g* at 4 °C for 10 min and 150 µL of dimethyl sulfoxide (DMSO) (SRL chemicals, Mumbai, India) was added to solubilize the formazan crystals. The absorbance of each well was determined at 570 nm using a microplate reader (BioTeK, USA).

#### 2.4.3. Measurement of Cytokine

Macrophages were seeded at a concentration of 2 × 10^5^ cells/well in 96-well plates and cultured with different concentrations of LPS (0.001, 0.01, 0.1, 1, and 10 µg/mL) for 24 and 48 h. Macrophages were co-incubated with different concentrations of EPS400 and an optimized concentration of LPS (dissolved in RPMI) for 24 h. The supernatant was collected for profiling the cytokine levels (IL-6, IL-1β, TNF-α, and 1L-10) using commercial mouse ELISA kits (BT Bioassay, Shanghai, China), following the manufacturer’s instruction.

#### 2.4.4. Measurement of Nitrite Oxide

Peritoneal macrophages were seeded at a concentration of 2 × 10^5^ cells/well in 96-well plates and cultured with different concentrations of LPS (0.001, 0.01, 0.1, 1, and 10 µg/mL) for 24 and 48 h. Macrophages were co-incubated with different concentrations of EPS400 and optimized concentration of LPS for 24 h at 37 °C in a 5% CO_2_ humidified incubator. The NO level in cell culture supernatant was determined using the Griess reaction [23]. In brief, the cell supernatant was mixed with an equal volume (100 μL) of Griess reagent (Sigma-Aldrich, Rockville, USA) and incubated at room temperature for 45 min, before taking the absorbance at 550 nm using a microplate reader (BioTeK, USA). A sodium nitrite standard curve was used to calculate the amount of nitrite in the sample.

#### 2.4.5. Phagocytosis Activity

The effect of EPS400 on the phagocytosis of peritoneal macrophages was determined following the method given by [20]. Briefly, peritoneal macrophages were seeded at a concentration of 10^5^ cells/ mL to a cell culture dish (Himedia, India) and incubated at 37 °C for 2 h. Then, the macrophages were cultured with different concentrations of EPS400 and incubated at 37 °C for 30 min. After that, an aliquot of 100 µL yeast suspension (10^8^ cells/mL) was added to the cell culture dish and incubated at 37 °C for 1 h in 5% CO_2_ incubator. The cell culture dish was stained with May–Grünwald dye (Himedia, India) after being treated with 1 mL of 1% (*w*/*v*) tannic acid solution for 1 min. The cells were then rinsed in buffer and stained for 15 min with freshly diluted Giemsa solution (Himedia, India). Finally, the cells were examined under a light microscope at a magnification of 1000×. The percent of phagocytosis was calculated by taking into account the number of macrophages with internalized yeast per hundred macrophages.

### 2.5. Statistical Analysis

Experiments were performed in triplicate and the results were expressed as mean ± standard error mean (SEM). GraphPad Prism (version 5.01, USA) was applied for Analysis of Variance (ANOVA: one way or two way) wherever appropriate, a post-test using the Bonferroni method was used, and * *p* < 0.05; **, *p* < 0.01; ***, *p* < 0.001 as well as ^a–c^ *p* < 0.05 were employed to indicate significant differences.

## 3. Results and Discussion

In our previous research work, we achieved success in the isolation of numerous probiotic strains derived from fermented dairy products and infant feces. These investigations unequivocally showcased the formidable capabilities of the NCDC400 strain, particularly in its capacity for EPS production and its potential application in the domain of functional food development [11]. With the objective of substantiating this hypothesis, the current study undertook the isolation of EPS from the NCDC400 strain and subsequently subjected it to a comprehensive assessment encompassing a diverse array of antioxidative and immunomodulatory characteristics.

### 3.1. Yield of EPS400

The yield of EPS400 produced by *L. fermentum* NCDC400 in an optimized deproteinized whey medium was found to be 2.6 ± 56 g/L on wet basis, 290 ± 11.25 mg/L on dry basis, and 0.721 ± 0.25 mg/mg of bacterial cell on dry basis. These findings are consistent with previous research, in which we discovered a similar yield of another exopolysaccharide EPSRam12 derived from native bacterial strain *Lacticaseibacillus rhamnosus* Ram12 with quantification values of 2.1 g/L on a wet basis, 253 ± 13.5 mg/L on a dry basis, and 0.76 ± 0.21 mg/mg of the bacterial cell on dry basis [6]. EPS from *Lactobacillus plantarum* R301 has been reported to yield 97.85 mg/mL under the optimized conditions [10]. Disparities in EPS production yield can be ascribed to a multitude of factors, encompassing bacterial strains, nutrient composition, temperature, pH, and extraction method, among others [24].

### 3.2. Antioxidative Activities of EPS400

EPS400, at all concentrations (0.125, 0.25, 0.5, 1.0, 2.0, 3.0, 4.0, and 5.0 mg/mL), exhibited clear scavenging activities against the radicals in a concentration-dependent manner (Figure 1). However, at EPS concentrations greater than 3.0 mg/mL, it showed a slight increase in scavenging capacity. This could be attributed to the decreased solubility of EPS after a certain concentration [25].

DPPH is a stable free radical, and its scavenging activity can be used as a benchmark to assess the potential of an antioxidant [17]. DPPH radicals are frequently employed to assess a natural compound’s capacity to neutralize free radicals. As shown in Figure 1A, EPS400 showed a clear scavenging effect on DPPH radicals in a concentration-dependent manner. A scavenging ability of 71.17% was observed at 3.0 mg/mL concentration of EPS400 against DPPH free radicals, which is significantly (*p* < 0.05) less than VC (94.13%). EPS400 may denote an electron or hydrogen transforming DPPH radicals into a stable non-radical form (DPPH-H), declining the absorbance of liquid with color change from deep violet to yellow or even colorless [26]. Previously, it was documented that EPS originating from *Agrocybe cylindracea* exhibited a noteworthy capacity for scavenging DPPH radicals [26]. However, in contrast, EPS generated by *Lactobacillus helveticus* KLDS1 and *Lactiplantibacillus plantarum* YW11 demonstrated a relatively modest DPPH-scavenging ability [17,25].

EPS400 had an obvious scavenging effect on ABTS free radicals (Figure 1B) and showed a concentration-dependent relationship in the concentration range of 0.125–5 mg/mL. At the concentration of 3 mg/mL, the scavenging rate of EPS400 was 84.92%, which is significantly (*p* < 0.05) lower than the positive control VC (94.91%). Aligned with this observation, Wang et al. (2018) conducted an evaluation of the scavenging rates exhibited by diverse molecular weight variants of Astragalus polysaccharide. Their findings unveiled a dose-dependent efficacy in scavenging ABTS free radicals. Interestingly, the study highlighted that those polysaccharides of moderate molecular weight demonstrated the most pronounced antioxidant activity, albeit slightly below the effectiveness of VC [27].

Similarly, the scavenging rates on hydroxyl radical of EPS400 (63.92%) were lower than those of the positive control VC (94.34%) at 3 mg/mL (Figure 1C). The hydroxyl ion, recognized as the most potent reactive radical among the spectrum of reactive oxygen species, possesses the capability to breach cell membranes and engage with biomolecules, thus instigating the formation of detrimental compounds that induce damage to tissues and cells [26]. The efficacy of EPS400 in scavenging hydroxyl radicals can potentially be attributed to its capacity for ion chelation, which in turn can thwart the generation of hydroxyl radicals [26]. Analogous findings concerning the scavenging potential of EPS derived from *Lactobacillus helveticus* KLDS1 and *Lactiplantibacillus plantarum* YW11, targeting hydroxyl radicals, have been documented [17,25].

The rate of scavenging activities of EPS400 and VC on superoxide radicals was directly proportional to their concentrations (Figure 1D). The scavenging rate of EPS400 (65.89%) was lesser than VC’s (94.99%) at a 3 mg/mL concentration. In congruence with these findings, the scavenging potential of EPS generated by *Lactobacillus helveticus* KLDS1 and *Lactiplantibacillus plantarum* against the superoxide anion radical has been documented [17,25]. The superoxide anion radical assumes the role of a precursor to lipid peroxidants, including hydrogen peroxide, singular oxygen, and hydroxyl radicals. These entities engage with biological molecules such as lipids, proteins, and DNA, thereby instigating cellular damage and contributing to various diseases [17,25]. Thus, the scavenging efficacy of EPS400 against the superoxide radical holds potential in mitigating oxidative stress.

The ability to chelate metals is widely regarded as an antioxidant mechanism due to its impact in diminishing the concentration of catalytic transition metals that participate in lipid peroxidation [17]. Within the spectrum of transition metals, the ferrous ion stands out as a particularly reactive prooxidant that plays a role in generating hydroxyl radicals [25]. The act of chelating ferrous ions is viewed as a complementary method for evaluating the antioxidant efficacy of a compound. This mechanism operates in conjunction with the direct scavenging of hydroxyl radicals [25]. EPS400 and EDTA chelate ferrous ions in a concentration-dependent manner (Figure 1E). The ferrous chelating ability of EPS400 was 66.63%, which is weaker than that of EDTA at a 3.0 mg/mL concentration, but more than half of the ability of EDTA (92.99%). Regarding the metal-ion-chelating ability of LAB-derived EPS, it has been reported that EPS samples on ferrous ion were weaker than that of EDTA-Na [17], the ferrous chelating rate of the EPS from *L. plantarum* YW11 increased with the increase in concentration, and the highest chelating rate (41.09%) achieved was close to half of the ability of EDTA (86.25%) [25].

The capacity of EPS sourced from microbial origins to donate hydrogen has been identified as a prominent attribute contributing to specific antioxidant functions. This mechanism bears resemblance to the functioning of polysaccharides derived from alternative sources [28]. A uniform approach has been devised for their application, encompassing the enhancement of EPS structures through the incorporation of diverse functional groups that either donate or withdraw electrons. Additionally, the integration of monosaccharides rich in reducing sugars, notably those featuring aldehyde or ketone groups, has been employed to reinforce this approach [28]. The presence of functional groups such as carboxylates (C=O) in EPS400 might be responsible for the binding of divalent cations and their antioxidant activities. The antioxidant activities of polysaccharides appear to be caused by a combination of factors, including but not limited to, exopolysaccharide structural features such as monosaccharide residues, branching, molecular weight, glycosidic linkage, functional groups, and chemical modifications [17,28].

### 3.3. Antimicrobial Activity of EPS400

EPS400 showed noteworthy antipathogenic activities against various food spoilage bacterial pathogens such as *Pseudomonas aeruginosa* NCDC105, *Micrococcus luteus* NCDC174, *Escherichia coli, Staphylococcus aureus* NCDC100, and *Bacillus substilis* NCDC70. The results showed that EPS400 generated very strong inhibition zones (≥15 mm) against *Pseudomonas aeruginosa* NCDC105 (18 ± 0.7 mm), and *Micrococcus luteus* NCDC174 (16 ± 0.57 mm), and a strong zone of inhibition (11–14 mm) against *Escherichia coli* NCDC135 (13 ± 0.57 mm), *Staphylococcus aureus* NCDC100 (11.3 ± 0.53 mm), and *Bacillus substilis* NCDC70 (11.3 ± 0.33 mm). However, no zone of inhibition against fungus, *Rhodotorula glutinis* NCDC51, *Aspergillus niger* NCDC55, *Candida butyri* NCDC280, and *Penicillum camemberti* NCDC56 was observed.

Some studies have shown that EPS from different LAB has varying degrees of antimicrobial activity against a variety of pathogens in vitro and have proposed several possible antibacterial mechanisms of EPS, including (1) the chelation of metal, which causes the suppression of nutrients essential for microbial growth, (2) interaction with the ionic surface, which disrupts the cell wall and cytoplasmic membrane, and (3) the stimulation of different TLR-mediated signaling processes, which causes the inhibition of mRNA and proteins synthesis [4,29,30,31]. A significant antibacterial activity of EPS-Ca6 produced by *Lactobacillus* sp. Ca6 strain against Micrococcus luteus and Salmonella enteric with 14 and 10 mm of inhibition zone, respectively, was observed by [31]. EPS-C70 derived from Lactiplantibacillus plantarum reduced food pathogens such as *E. coli* and *S. aureus* by 2 to 3 logs, as reported by [4]. Similarly, the antibacterial activities of B-EPS and L-EPS produced by Bifidobacterium bifidum WBIN03 and Lactiplantibacillus plantarum R315, respectively, against *Cronobacter sakazakii*, *Escherichia coli*, *Listeria monocytogenes*, *Staphyloccocus aureus*, *Candida albicans*, *Bacillus cereus, Salmonella typhimurium*, and *Shigella sonnei* was reported by [29]. The potential antibacterial activity of EPS400 can be attributed to its high molecular weight and the presence of several functional groups, such as carbonyl and hydroxyl groups, which have been suggested to play an important role in employing the antimicrobial effects of EPS [4]. However, EPS400 exhibited no antifungal activity, which could be attributed to the difference in the nature of fungus and bacteria cell walls.

### 3.4. Immunomodulatory Potential of EPS400 in Murine Peritoneal Macrophages

#### 3.4.1. Optimization of Concentration of LPS

It is well known that LPS can bind to Toll-like receptor 4 (TLR4) and activate the immune system by inducing cytokine release [23]. Thus, LPS-activated macrophages may be the best model for studying the immunomodulatory effects of test compounds, as well as the dynamics of the release of a variety of pro-inflammatory cytokines. Therefore, the different concentrations of LPS (0.001, 0.01, 0.1, 1, and 10 µg/mL) and times of 24 to 48 h were optimized to find out the ideal acting condition for evaluating the anti-inflammatory effect of EPS400 in murine peritoneal macrophages (Figure 2). MTT assay showed that LPS had no significant cytotoxic effects at a concentration up to (0.01 µg/mL) 10 ng/mL at 24 h as compared with control cells that received no treatment. Cell viability began to decrease after 0.1 µg/mL and decreased below 50% when the LPS concentration was increased to 10 µg/mL at 24 h (Figure 2A). Previously, MTT assays revealed that LPS had a significant inhibitory effect on RAW macrophage survival after a 72 h treatment at 100–1000 ng/mL [32].

The production of the anti-inflammatory cytokine IL-10 increased significantly (*p* < 0.01) with the treatment of LPS for 24 or 48 h, up to 0.01 µg/mL, and decreased significantly (*p* < 0.01) from 1 µg/mL to 10 µg/mL, compared to the control (Figure 2B). A non-significant difference was observed at 0.1 µg/mL in the treatment of LPS for 24 h, compared to the control. A significant increase (*p* < 0.001) in the production of IL-6 (Figure 2C), TNF-α (Figure 2D), and NO (Figure 2E) was observed from 0.01 µg/mL after 24 h and 48 h of LPS treatment compared to the control (Figure 2). At a 0.001 µg/mL concentration of LPS, a significant increase (*p* < 0.001) in the production of NO after 48 h of LPS treatment was found, while a non-significant difference was observed after 24 h compared to the control. The highest increase in NO was observed at a 1µg/mL concentration of LPS treatment after 24 h and 48 h, compared to the control, with no remarkable change between 24 h and 48 h (Figure 2E). A significant increase (*p* < 0.001) in the production of NO, IL-6, and TNF-α, and a decrease in anti-inflammatory cytokine IL-10 (*p* < 0.01) were observed at 1 µg/mL concentration of LPS treatment after 24 h, compared to the control, whereas no significant difference between 1 µg/mL and 10 µg/mL concentrations of LPS treatment for 24 h and 48 h was observed. Therefore, the concentration of 1 µg/mL of LPS for 24 h was selected for the challenging the murine macrophage to evaluate the anti-inflammatory effect of EPS400.

#### 3.4.2. Effect of EPS400 on Cell Viability

Preceding the exploration of EPS400′s anti-inflammatory potential, an MTT assay was performed to assess its cytotoxicity towards murine macrophages (Figure 3A). As depicted in Figure 3A, treatment with EPS400 at concentrations spanning 12.5 to 800 μg/mL did not induce a substantial decrease (*p* > 0.05) in the viability of macrophage cells compared to the control. Contrarily, a noteworthy elevation in murine macrophage viability was evident at EPS400 concentrations of 12.5, 25, and 50 μg/mL relative to the control group (*p* < 0.05). This observed enhancement in cell viability could be attributed to the availability of vital nutrients, such as sugars, essential for cellular growth. This outcome resonates with the findings of [22], wherein no decline in cell viability of RAW264.7 macrophages was noted following treatment with acidic EPS from *Lactiplantibacillus plantarum* JLAU103 within the concentration range of 20–100 μg/mL. Moreover, ref. [22] reported a significant augmentation in cell viability at concentrations of 40, 60, and 80 μg/mL, in comparison to the control group. Subsequently, EPS400 concentrations of 12.5, 25, and 50 μg/mL, which exhibited an augmented cell viability, were chosen for further investigation.

#### 3.4.3. Anti-Inflammatory Effect of EPS400 in Murine Peritoneal Macrophages

An excessive and dysregulated production of cytokines poses a significant threat, contributing to a range of inflammatory disorders [23]. Consequently, an approach focused on regulating the release of pro-inflammatory cytokines emerges as a promising strategy for the development of anti-inflammatory medications. In this context, the impact of EPS400 on cytokine secretion by mouse macrophages was assessed, particularly concerning alterations in pro-inflammatory cytokines such as IL-6, IL-1β, and TNF-α, as well as the anti-inflammatory cytokine IL-10, within a LPS-treated primary macrophage cell line. It was observed that the activation of macrophages by LPS (1 μg/mL) significantly augmented the release of pro-inflammatory cytokines, namely IL-6, IL-1β, and TNF-α, in comparison to the control group (*p* < 0.001). Treatment with different concentrations of EPS400 significantly decreased (*p* < 0.05) the pro-inflammatory cytokine, including IL-6 (Figure 3B), IL-1β (Figure 3C), and TNF-α (Figure 3D), and increased the anti-inflammatory cytokine IL-10 (Figure 3E) in LPS-activated mouse macrophages in a concentration-dependent manner. A reduction in the level of IL-6 of 68.1%, 77.01%, and 85.56%, IL-1β of 51.7%, 65.3%, and 78.2%, and TNF-α of 55.5%, 76.7%, and 84.4%, while increases in the level of IL-10 of 139.6%, 200.3%, and 281.4%, with the treatment of EPS400 were observed at the concentrations 12.5, 25, and 50 μg/mL, respectively, compared to the LPS-alone-treated cell (Figure 3B–E).

LPS, an endotoxin of bacterial origin, triggers the release of pro-inflammatory cytokines along with other molecules such as nitric oxide (NO). Following LPS stimulation, the concentration of NO in the culture supernatant of macrophage cells exhibited a marked increase compared to untreated control cells, as illustrated in Figure 3F. In alignment with this discovery, ref. [33] also identified a substantial surge in NO production within the culture supernatant following LPS stimulation of RAW 264.7 macrophages, in contrast to the untreated control group. Nitric oxide (NO) assumes a pivotal role in the initiation of inflammation. While it exerts anti-inflammatory effects under normal physiological conditions, NO can function as a pro-inflammatory mediator due to its excessive production in specific abnormal scenarios [22,23]. Consequently, inhibitors targeting NO might offer assistance in the mitigation of inflammatory disorders. The treatment with EPS400 significantly reduced the LPS-stimulated macrophage’s NO generation in a dose-dependent manner (*p* < 0.001). EPS400 treatment resulted in 26.8%, 45.5%, and 60.1% decrease in NO production at the concentrations of 12.5, 25, and 50 μg/mL, compared to the LPS-alone-treated cell. In line with this result, polysaccharides derived from *Cyclocarya paliurus* have been reported to suppress the production of NO in a dose-dependent manner in LPS-treated macrophage cells [34]. Similarly, ref. [22] reported a significant reduction in NO production after treatment with acidic EPS produced by *L. planetarium* JLAU103 at concentrations of 40, 60, and 80 g/mL in comparison to the LPS-alone-treated cell group.

The immunomodulatory attributes exhibited by EPS can be attributed to factors encompassing its molecular weight, structure, and composition, which collectively influence the nature of the immune response generated. The interplay between EPS’s molecular weight and its potential to counteract inflammation has been comprehensively elucidated [35]. Notably, ref. [35] demonstrated that EPSs characterized by high molecular weights are capable of initiating the activation and differentiation of dendritic cells (DCs). This process culminates in the production of cytokines and subsequent orchestration of the differentiation of naive T cells into regulatory T cells (Tregs), a phenomenon that effectively governs an excessive T cell response. Moreover, EPS exhibits the capacity to stimulate macrophages, triggering the secretion of cytokines that potentially foster homeostasis through the reduction of pro-inflammatory cytokine levels. The inclusion of unique sugars such as L-rhamnose within polysaccharides has been reported across various applications, spanning from anti-inflammatory and antioxidant agents to the synthesis of nucleoside analogs employed as antiviral agents [36]. Drawing from the insights provided by the aforementioned study, the anti-inflammatory qualities of EPS400 can likely be ascribed to its heteropolysaccharide constitution, specifically the presence of L-rhamnose, as well as its substantial molecular weight.

#### 3.4.4. Effect of EPS400 on Phagocytosis Activity in Murine Peritoneal Macrophages

Phagocytosis, as a fundamental cellular process, plays an important role in host defense. Phagocytic activity is a commonly used method for assessing the activities of macrophage function modulators. Hence, the biological effects of EPS400 on the phagocytic activity of macrophages were investigated and found to be positive, as shown in Figure 4. The number of yeast-engulfed macrophages followed a linear trend, indicating that EPS400 enhanced phagocytic response in a dose-dependent manner. Compared with the control group, the phagocytic activities of macrophages treated at the concentrations of 12.5, 25, and 50 μg/mL were increased significantly, being 33.5%, 43.8%, and 58.6%, with EPS400, respectively (Figure 4). Our results, in this regard, are in close agreement with the previous reports of the phagocytic ability of EPS from LAB strains. EPS103 derived from *L. plantarum* JLAU103 [22] and R-5-EPS produced from *L. helveticus* LZR-5 [37] have been reported to enhance phagocytic activity.

While the exact mechanism through which EPS influences the regulation of macrophage phagocytic activity remains elusive, it is evident that phagocytosis is triggered through various receptor–ligand interactions, leading to the elimination of pathogens and deceased cells within the host organism [38,39,40,41] (Figure 5). This process involves the engagement of different Toll-like receptor (TLR)-mediated signaling pathways, which in turn play a role in regulating pathogen phagocytosis by macrophages. Part of this regulation can be attributed to the involvement of signaling molecules that interact with distinct TLRs [30].

## 4. Conclusions

In this study, an extensive exploration into the biological properties of EPS400, derived from *L. fermentum* NCDC400, has provided insights into its multifaceted potential encompassing antibacterial, antioxidant, and immunomodulatory activities. The findings presented herein underscore the noteworthy contributions of EPS400, notably its robust scavenging activity in targeting diverse radicals, as well as its remarkable antibacterial efficacy against prevalent food spoilage bacteria. Notably, EPS400 emerged as a regulator of cytokine release, effectively curbing the excessive secretion of TNF-α, IL-1β, IL-6, and NO, while concurrently fostering the release of the anti-inflammatory cytokine IL-10. Furthermore, EPS400 emerged as a potent enhancer of the phagocytic capability of macrophages. The cumulative outcomes of this investigation accentuate the promising prospects of EPS400 as a prospective bioactive constituent for integration into functional foods or medicinal formulations. However, it is prudent to acknowledge that further validation through in vivo research is imperative. This validation process should focus on corroborating the antioxidant, antibacterial, and immunomodulatory attributes of EPS400, particularly with an emphasis on unraveling its intricate structural features. In conclusion, this study lays the groundwork for potential applications of EPS400 in promoting health and combating ailments, while underscoring the necessity for continued research to realize its full therapeutic potential.

## Figures and Tables

**Figure 1 foods-12-03595-f001:**
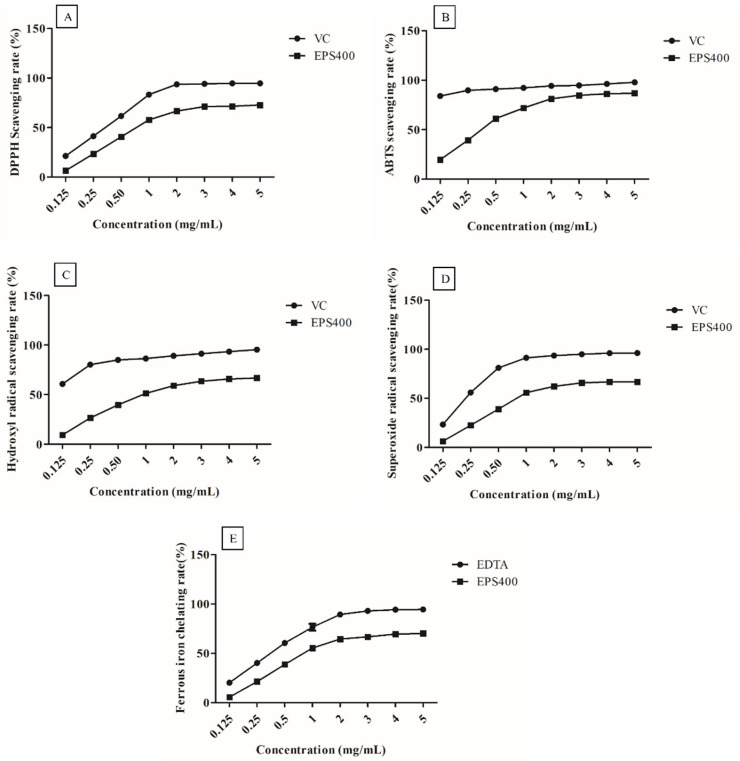
Scavenging activities of the EPS400 and vitamin C (VC) against: (**A**) 2-diphenyl-1-picrylhydrazyl (DPPH) radicals, (**B**) 2,2′-azino-di-(3-ethylbenzthiazoline sulfonic acid) (ABTS) radical, (**C**) hydroxyl radicals, and (**D**) superoxide anion, as well as (**E**) Fe^2+^ chelating ability of EPS400 and Ethylenediaminetetraacetic acid (EDTA). Points denote the mean values of scavenging or chelating rate.

**Figure 2 foods-12-03595-f002:**
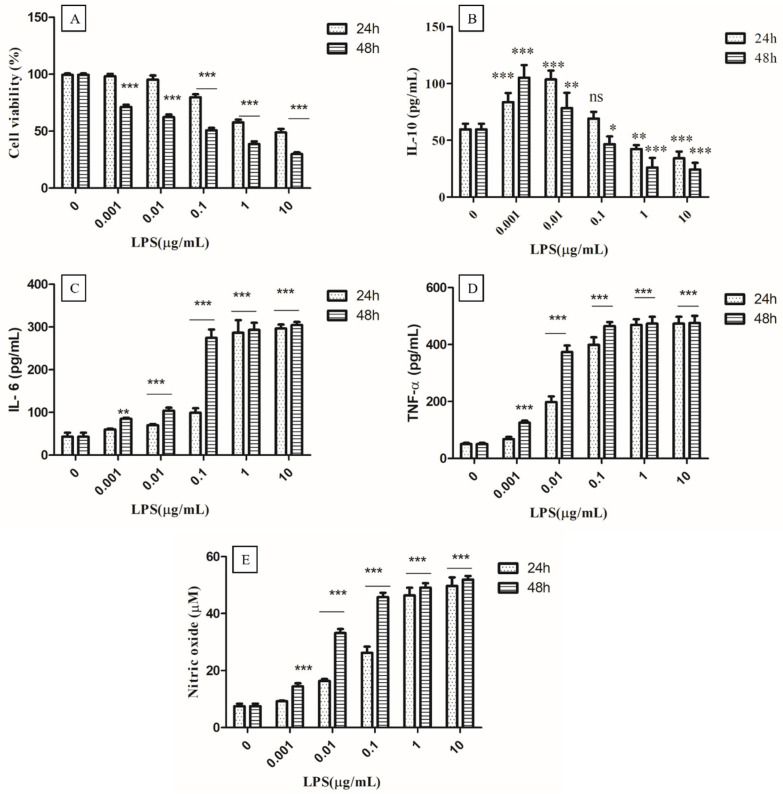
Optimization of concentration of LPS (Lipopolysaccharides). (**A**) Cell viability, (**B**) Anti-inflammatory cytokine IL-10, (**C**) Proinflammatory IL-6, and (**D**) TNF-α, as well as (**E**) nitric oxide. Cells cultured in the serum-free medium without any treatment were used as control. Data are presented as the mean ± SEM (*n* = 3) (* *p* < 0.05; **, *p* < 0.01; ***, *p* < 0.001 vs. control group).

**Figure 3 foods-12-03595-f003:**
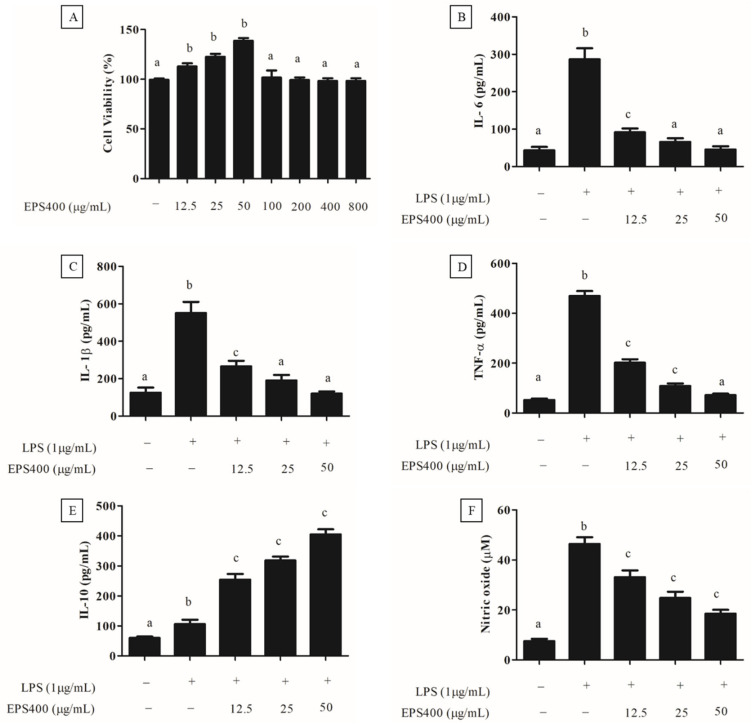
Effect of EPS400 on murine macrophages. (**A**) cell viability, (**B**) cytokine IL-6, (**C**) IL-1β, (**D**) TNF-α, (**E**) IL-10, and (**F**) nitric oxide. Cells cultured in the serum-free medium without any treatment were used as control. Data are presented as the mean ± SEM (*n* = 3). Superscript a–c (*p* < 0.05) compared with control and LPS group.

**Figure 4 foods-12-03595-f004:**
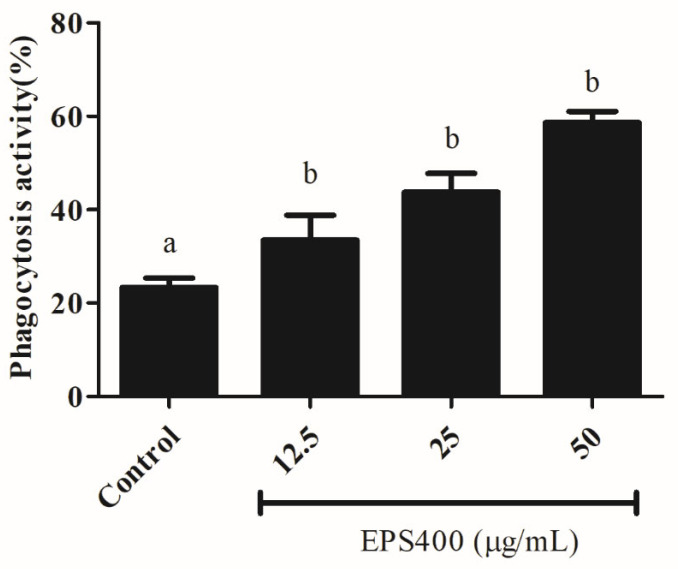
Effect of EPS400 on phagocytic activity. Cells cultured in the serum-free medium without any treatment were used as control. Data are presented as the mean ± SEM (*n* = 3). Superscript a, b (*p* < 0.05) compared with control and LPS group.

**Figure 5 foods-12-03595-f005:**
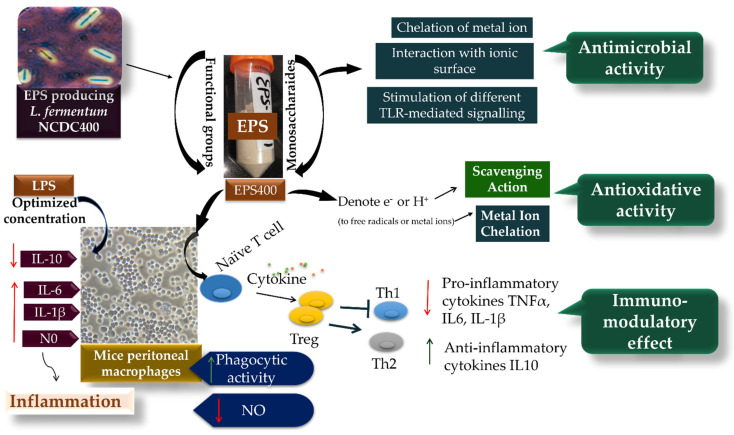
The potential mechanism of EPS400 in exhibiting phagocytic, anti-microbial, and anti-oxidative activities.

## Data Availability

The data used to support the findings of this study can be made available by the corresponding author upon request.

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
