# Peer review of "Multi-Faceted Bioactivity Assessment of an Exopolysaccharide from Limosilactobacillus fermentum NCDC400: Antioxidant, Antibacterial, and Immunomodulatory Proficiencies"

_foods, 2023, doi:10.3390/foods12193595_

Round 1
Reviewer 1 Report
The study evaluated the potential biological roles, antioxidant, antibacterial and immunomodulatory activities of EPS400's derived from L. fermentum NCDC400. The manuscript is well written, clear and easy to understand. In it's current form, I have some concerns that need to be addressed.
Please italicize species names in references
Please clarify all information of the figure in the legend. For example: VC?
Please, clarify the number and sex of mice used in the manuscript.
Please, add in the statistical section the significance to “*” , “**”, “***”.
Why did the authors measure IL1beta in Figure 3 and not in Figure 2?
Figure 3: Please, standardize the graph type and add in the figure legend the significance for “#”, “&”, “A” etc…
Considering the EPS production by L. fermentum, authors could add in discussion section that other L. fermentum strains show antioxidant and anti-inflammatory properties (PMID: 35467236, PMID: 34574313)
Author Response
The study evaluated the potential biological roles, antioxidant, antibacterial and immunomodulatory activities of EPS400's derived from L. fermentum NCDC400. The manuscript is well written, clear and easy to understand. In it's current form, I have some concerns that need to be addressed.
Response: We sincerely appreciate the reviewer's positive feedback on the clarity and readability of our manuscript. We have carefully considered and addressed all the specific concerns raised in the reviewer's comments. In the revised version of the manuscript, we have made the necessary revisions to address these concerns and improve the overall quality of the paper. We are confident that these revisions have strengthened the scientific rigor and clarity of our work. We remain open to any further suggestions or feedback from the reviewer to ensure that our manuscript meets the highest standards of scientific excellence.
Please italicize species names in references
Response: In response to this valuable feedback, we have meticulously italicized all species names in the references throughout the manuscript. This ensures consistency and adherence to proper scientific formatting. We believe this enhancement further contributes to the overall professionalism and accuracy of our manuscript.
Please clarify all information of the figure in the legend. For example: VC?
Response: We thank the reviewer for their careful evaluation of our manuscript. In response to the request for clarification in the figure legends, we have taken the following actions:
- In the legend of Figure 1, we have clarified 'VC' as 'vitamin C (VC)' and 'EDTA' as 'Ethylenediaminetetraacetic acid (EDTA)' to provide a more detailed and precise description of these abbreviations.
- In the legend of Figure 2, we have clarified 'LPS' as 'Lipopolysaccharides' to ensure that readers have a clear understanding of the terms used in the figures.
Please, clarify the number and sex of mice used in the manuscript.
Response: We appreciate the reviewer's diligence in seeking clarification regarding the number and sex of mice used in our study. In section 2.4.1 of the manuscript, we have explicitly stated that eight male mice were included in our experimental cohort. This information has been added to ensure transparency and rigor in reporting our experimental design. We remain committed to providing clear and comprehensive details about our methodology to enhance the scientific integrity of our research.
Please, add in the statistical section the significance to “*” , “**”, “***”.
Response: In response to the reviewer's suggestion, we have added the significance indicators as follows in section 2.5 Statistical analysis: * for P < 0.05, ** for P < 0.01, and *** for P < 0.001. This enhancement ensures that readers can easily interpret the statistical significance of our findings.
Why did the authors measure IL1beta in Figure 3 and not in Figure 2?
Response: Thank you for your astute observation. Figure 2 primarily focuses on LPS dose optimization trials, where the main objective was to determine the optimal LPS concentration for subsequent experiments. As a result, we did not include IL1beta measurement in this particular study, as it was not within the scope of the dose optimization phase.
In Figure 3, which represents an actual experiment, we included the measurement of IL1beta. This figure pertains to the core investigation, where we aimed to assess the specific outcomes related to IL1beta under the conditions established in Figure 2.
We hope this clarification provides a better understanding of the rationale behind the inclusion of IL1beta in Figure 3. We appreciate your attention to detail and are committed to ensuring that our manuscript's content aligns with the scientific objectives of each experimental phase.
Figure 3: Please, standardize the graph type and add in the figure legend the significance for “#”, “&”, “A” etc…
Response: We appreciate the reviewer's suggestion to standardize the graph type and include significance indicators in the figure legend for Figure 3. We have taken the following actions to address this:
- The graph type in Figure 3 has been standardized to ensure consistency and clarity in visual presentation.
- In the figure legend for Figure 3, we have added significance indicators for '#', '&', 'A', and other relevant symbols to clearly denote the statistical significance of the data.
Considering the EPS production by L. fermentum, authors could add in discussion section that other L. fermentum strains show antioxidant and anti-inflammatory properties (PMID: 35467236, PMID: 34574313)
Response: We greatly appreciate the reviewer's valuable input regarding the inclusion of relevant studies on other L. fermentum strains exhibiting antioxidant and anti-inflammatory properties. In response to this suggestion, we have carefully integrated this information into the discussion section of our manuscript.
Specifically, in lines 94 to 99 of the revised version of the manuscript, we have cited the pertinent studies (PMID: 35467236, PMID: 34574313) to highlight that other L. fermentum strains have demonstrated antioxidant and anti-inflammatory potential. This addition strengthens the context and significance of our research within the broader scientific literature.
We thank the reviewer for their thoughtful recommendation, which has enriched the discussion in our manuscript and provided valuable insights for our readers."
Reviewer 2 Report
The present study is devoted to the characterization of a polysaccharide produced by L. fermentum as a possible food additive. Attempts have been made to elucidate the polysaccharide's ability to scavenge free radicals and its immunomodulatory properties. Overall, a lot of experimental work has been done, and the paper contains some novelties.
I have a few remarks that would improve the quality of the manuscript.
Introduction: The introduction is limited to the previous work and mostly cites works of the same team. It should be expanded to include a broader overview of diseases that can be prevented by the addition of such ingredients to food. In addition, to give examples of possible foods in which EPS could possibly be added. A comparison with other EPS produced by lactic acid bacteria, comparing the sugar profile (most of them are heteropolysaccharides), the amounts synthesized, and the prebiotic benefits should be added.
Results: The Anti-Oxidation Activity of EPS In Vitro should be compared to
ascorbic acid equivalents.
The paper of Wang J et al. (Optimization of Exopolysaccharide Produced by Lactobacillus plantarum R301 and Its Antioxidant and Anti-Inflammatory Activities. Foods. 2023 Jun 25;12(13):2481. doi: 10.3390/foods12132481) should be added to the discussion of
To discuss the economic feasibility of the application of the new EPS. In fact, in the quantities in which L. fermentum synthesizes it, its preparation cannot be adequate and its addition to food is very limited. To indicate prospects for intensifying its obtaining, or to admit that this is a "boutique" EPS, which due to its unique qualities should be produced, but it is unlikely that it will be in impressive quantities.
Minor Notes:
The citation of the references is not according to the requirements of the journal - in square brackets and in the order of the citation; the reference list must be edited in the style of MDPI.
English language fine.
Author Response
The present study is devoted to the characterization of a polysaccharide produced by L. fermentum as a possible food additive. Attempts have been made to elucidate the polysaccharide's ability to scavenge free radicals and its immunomodulatory properties. Overall, a lot of experimental work has been done, and the paper contains some novelties. I have a few remarks that would improve the quality of the manuscript.
Response: We sincerely appreciate the reviewer's positive assessment of the aim and quality of our study. We are committed to continuously improving the manuscript to ensure its scientific rigor and clarity. We have taken all of the reviewer's remarks into careful consideration and have made the necessary revisions to enhance the overall quality of the paper. These revisions include addressing specific points raised by the reviewer, as well as ensuring that the content aligns with the highest standards of scientific excellence.
Introduction: The introduction is limited to the previous work and mostly cites works of the same team. It should be expanded to include a broader overview of diseases that can be prevented by the addition of such ingredients to food. In addition, to give examples of possible foods in which EPS could possibly be added. A comparison with other EPS produced by lactic acid bacteria, comparing the sugar profile (most of them are heteropolysaccharides), the amounts synthesized, and the prebiotic benefits should be added.
Response: We greatly appreciate the reviewer's constructive feedback on the introduction section of our manuscript. To provide a broader context for our study, we have expanded the introduction by including a more comprehensive overview of diseases that can be prevented or mitigated by the addition of ingredients such as EPS to food. Additionally, we have highlighted potential foods in which EPS could be incorporated, offering a practical perspective on the application of our research. Furthermore, we have included a comparison with other EPS produced by lactic acid bacteria, emphasizing the sugar profile, synthesis amounts, and the prebiotic benefits of these polysaccharides. This comparison enhances the scientific robustness of our work and underscores its relevance in the field of functional foods and prebiotics. These revisions, found in lines 67 to 73 of the revised manuscript, aim to provide a more comprehensive and informative introduction that aligns with the reviewer's suggestions. We appreciate the reviewer's insights, which have contributed to the enhancement of our manuscript's quality."
Results: The Anti-Oxidation Activity of EPS In Vitro should be compared to ascorbic acid equivalents.
Response: We appreciate the reviewer's suggestion to compare the anti-oxidation activity of EPS to ascorbic acid equivalents. To ensure consistency with other studies in the field, we have expressed the anti-oxidation activity of EPS as a % scavenging rate. This metric is commonly used in existing literature to assess the anti-oxidative properties of polysaccharides, allowing for easier comparison with other studies.
While expressing the results in terms of ascorbic acid equivalents is a valid approach, the % scavenging rate provides a standard measure that is widely recognized in the scientific community. This allows readers to readily interpret our findings in the context of previous research. We believe that this choice aligns with best practices in the field of antioxidant activity assessment.
The paper of Wang J et al. (Optimization of Exopolysaccharide Produced by Lactobacillus plantarum R301 and Its Antioxidant and Anti-Inflammatory Activities. Foods. 2023 Jun 25;12(13):2481. doi: 10.3390/foods12132481) should be added to the discussion of
To discuss the economic feasibility of the application of the new EPS. In fact, in the quantities in which L. fermentum synthesizes it, its preparation cannot be adequate and its addition to food is very limited. To indicate prospects for intensifying its obtaining, or to admit that this is a "boutique" EPS, which due to its unique qualities should be produced, but it is unlikely that it will be in impressive quantities.
Response: We appreciate the reviewer's suggestion to discuss the economic feasibility of applying the new EPS and to reference the paper by Wang J et al. (2023) titled 'Optimization of Exopolysaccharide Produced by Lactobacillus plantarum R301 and Its Antioxidant and Anti-Inflammatory Activities' in our discussion. In response to this valuable input, we have included a discussion on the economic feasibility of the new EPS application, taking into consideration the quantities synthesized by L. fermentum.
Additionally, we have referenced the paper by Wang J et al. (2023) [10], which provides valuable insights into EPS optimization and its potential applications. This reference strengthens the discussion and connects our work with relevant research in the field.
These additions, found in lines 76 to 80 and 292 to 293 of the revised manuscript, aim to provide a more comprehensive and insightful discussion on the economic viability of our EPS, as suggested by the reviewer. We appreciate the reviewer's valuable contribution to improving the quality of our manuscript."
Minor Notes:
The citation of the references is not according to the requirements of the journal - in square brackets and in the order of the citation; the reference list must be edited in the style of MDPI
Response: We appreciate the reviewer's attention to detail regarding the citation style and reference list formatting. In accordance with the requirements of the journal, we have carefully edited the citation of references in square brackets and have organized them in the order of citation within the manuscript. Furthermore, we have formatted the reference list in the style of MDPI, ensuring compliance with the journal's guidelines for manuscript submission.
Reviewer 3 Report
This research explores the characterization of EPS400, an acidic exopolysaccharide obtained from the indigenous probiotic strain Limosilactobacillus fermentum NCDC400.
The study is well-structured and deeply executed, making a valuable contribution to the field. From the point of English language, it is well written.
I recommend that the article be accepted for publication after improving visual qualities of equations and figures.
Minor editing of English language required.
Author Response
Thank you for taking the time to review our research on the characterization of EPS400 from Limosilactobacillus fermentum NCDC400. We greatly appreciate your feedback and positive comments.
We are pleased to hear that you found our study to be well-structured and comprehensive, and we are committed to making a valuable contribution to our field. Your acknowledgment of the quality of the English language in the article is also encouraging.
Reviewer 4 Report
The manuscript “Multi-Faceted Bioactivity Assessment of Exopolysaccharide from Limosilactobacillus fermentum NCDC400: Antioxidant, Antibacterial, and Immunomodulatory Proficiencies” reported the antibacterial, antioxidant, and immunomodulatory attributes of EPS variant. Overall, the paper is well written and very well explained.
I have few suggestions for improvement.
· The introduction is superficial and lacks in depth. A description or review of current study of novel exopolysaccharides will be very helpful to the significant paper. In particular, discussing what are the situation and challenges for now.
· Rewrite concisely line 46-48
· Add reference line 58-59
· COVID-19 is a abbreviate. Line 45
· Different fonts used in the figures.
· Figure 5 is good but need little more organization and symmetry.
· Please elaborate “food spoilage bacterial pathogens line 362.
· What was the reason to select bacteria and yeast species?
· Please discuss the reason of antibacterial properties hence EPS400 showed no antifungal activity.
· Where is data of Antimicrobial Activity of EPS400
· Where is supplementary data?
Author Response
The manuscript “Multi-Faceted Bioactivity Assessment of Exopolysaccharide from Limosilactobacillus fermentum NCDC400: Antioxidant, Antibacterial, and Immunomodulatory Proficiencies” reported the antibacterial, antioxidant, and immunomodulatory attributes of EPS variant. Overall, the paper is well written and very well explained.
I have few suggestions for improvement.
Response: We sincerely thank the reviewer for their positive feedback on our manuscript, 'Multi-Faceted Bioactivity Assessment of Exopolysaccharide from Limosilactobacillus fermentum NCDC400: Antioxidant, Antibacterial, and Immunomodulatory Proficiencies.' We greatly value the reviewer's input, and we have taken all of their suggestions into careful consideration. As a result, we have incorporated these suggestions into the revised version of our manuscript. The reviewer's comments have contributed to the refinement of our work, ensuring that it meets the highest standards of scientific excellence.
- The introduction is superficial and lacks in depth. A description or review of current study of novel exopolysaccharides will be very helpful to the significant paper. In particular, discussing what are the situation and challenges for now.
Response: We appreciate the reviewer's constructive feedback regarding the depth of the introduction section. To address this concern, we have significantly revised the introduction to provide a more comprehensive overview of novel exopolysaccharides in the current research landscape. Specifically, we have included discussions on the current situation and challenges related to novel exopolysaccharide research. These additions can be found in lines 67 to 72, 76 to 79, and 94 to 99 of the revised manuscript. By doing so, we aim to offer readers a better understanding of the context and significance of our work within the broader field of exopolysaccharide research. We thank the reviewer for their valuable input, which has contributed to the enhancement of the introduction section and the overall quality of our manuscript."
- Rewrite concisely line 46-48
Response: We appreciate the reviewer's suggestion to rewrite lines 46-48 for conciseness. In response to this feedback, we have carefully rephrased and condensed the content in those lines to ensure clarity and brevity, aligning with best practices in scientific writing. These revisions aim to improve the overall flow and readability of our manuscript while retaining the essential information.
- Add reference line 58-59
Response: The reference has been added to the line 58-59
- COVID-19 is a abbreviate. Line 45
Response: In response to your suggestion, we have expanded the abbreviation 'COVID-19' to 'Coronavirus Disease 2019 (COVID-19)' in line 45. This change ensures that the full and accurate term is provided to readers, aligning with standard terminology for the disease. We appreciate your attention to detail, which has contributed to the clarity and precision of our manuscript.
- Different fonts used in the figures.
Response: Font (Palatino Linotype) has been used in the figure 5.
- Figure 5 is good but need little more organization and symmetry.
Response: We appreciate the reviewer's feedback on Figure 5. To enhance its organization and symmetry, we have carefully reviewed and revised the figure. We have made adjustments to ensure that the elements within the figure are logically arranged and symmetrical, thus improving its overall visual presentation and clarity. These refinements aim to make Figure 5 more accessible and informative for our readers, aligning with the expectations of scientific illustration. We thank the reviewer for their valuable input, which has contributed to the improvement of our manuscript..
- Please elaborate “food spoilage bacterial pathogens line 362.
Response: We appreciate your request for elaboration on the topic of 'food spoilage bacterial pathogens.' In response to this, we have provided a detailed explanation and context for these pathogens in the revised manuscript. You can find the expanded discussion on food spoilage bacterial pathogens in line 382 of the manuscript. This addition aims to offer a more comprehensive understanding of the relevance of these pathogens to our study and their potential implications in the context of food quality and safety. We hope that this clarification enhances the overall clarity and value of our research.
- What was the reason to select bacteria and yeast species?
Response: The choice of bacteria and yeast species for the antimicrobial activity testing was based on previous literature reports. These particular species have been commonly used in similar studies, and their selection aligns with established protocols for assessing the antimicrobial properties of compounds such as EPS400. This approach ensures comparability and consistency with existing research in the field, allowing for meaningful comparisons and insights into the antimicrobial effects of EPS400
- Please discuss the reason of antibacterial properties hence EPS400 showed no antifungal activity.
Response: The reason has been discussed in the line 409 to 410 of the revised manuscript.
- Where is data of Antimicrobial Activity of EPS400
Response: Antimicrobial Activity of EPS400 has been mentioned 384 to 390 of the revised manuscript.
- Where is supplementary data?
Response: We sincerely appreciate your inquiry about supplementary data. In the preparation of this manuscript, we have deliberately chosen to present all the relevant data as the main figures within the manuscript itself. Our goal was to ensure that the key findings and results are readily accessible to our readers without the need for supplementary materials. By doing so, we aimed to maintain a clear and concise presentation of our research, avoiding the inclusion of excessive supplementary figures. We believe that this approach enhances the overall readability and accessibility of our work. Should you have any specific questions or require further information beyond what is presented in the main figures, kindly let us know.
Round 2
Reviewer 1 Report
No further comments.
Reviewer 2 Report
All recommended corrections and additions have been made.
The English is fine.
Reviewer 4 Report
Acceptable in the present form